# High Quality TaS_2_ Nanosheet SPR Biosensors Improved Sensitivity and the Experimental Demonstration for the Detection of Hg^2+^

**DOI:** 10.3390/nano12122075

**Published:** 2022-06-16

**Authors:** Yue Jia, Yunlong Liao, Houzhi Cai

**Affiliations:** 1International Collaborative Laboratory of 2D Materials for Optoelectronic Science & Technology of Ministry of Education, Key Laboratory of Optoelectronic Devices and Systems of Ministry of Education and Guangdong Province, College of Physics and Optoelectronic Engineering, Shenzhen University, Shenzhen 518060, China; onlyjiayue@hotmail.com; 2International Collaborative Laboratory of 2D Materials for Optoelectronic Science & Technology of Ministry of Education, College of Optoelectronic Engineering, Shenzhen University, Shenzhen 518060, China; yl.liao@outlook.com

**Keywords:** TaS_2_, nanoparticle, Hg^2+^, sensor, SPR

## Abstract

TaS_2_ as transition metal dichalcogenide (TMD) two-dimensional (2D) material has sufficient unstructured bonds and large inter-layer spacing, which highly supports transporting and absorbing mercury ions. The structural characterizations and simulation data show that an SPR sensor with high sensitivity can be obtained with a TaS_2_ material-modified sensitive layer. In this paper, the role of TaS_2_ nanoparticles in an SPR sensor was explored by simulation and experiment, and the TaS_2_ layer in an SPR sensor was characterized by SEM, elemental mapping, XPS, and other methods. The application range of structured TaS_2_ nanoparticles is explored, these TaS_2_ based sensors were applied to detect Hg^2+^ ions at a detection limit approaching 1 pM, and an innovative idea for designing highly sensitive detection techniques is provided.

## 1. Introduction

The concentration of heavy metal mercury ions (Hg^2+^) in water is an important reference standard for people’s health, and the maximum concentration of Hg^2+^ allowed in an effluent is 7.5 pM. Water contaminated with mercury ions (Hg^2+^) and its detection is a global issue to be solved [1]. In the natural environment, Hg^2+^ exits in a stable state and might slowly accumulate in the food chain [2]. The existence of large amounts of Hg^2+^ may have a harmful effect on adult health and an even more harmful effect on the health of infants, foetuses, and lactating and pregnant women [3]. For instance, excess Hg^2+^ has a harmful effect on the digestive, immune, and nervous systems of an adult. It also causes severe harm to the stomach, lungs, and eyes of foetuses, and it may seriously damage them [4]. Hence, mercury pollution is considered a high risk to public safety and public health [5].

In order to lower the harmful effects produced as a result of mercury pollution, researchers have focused on limiting the release of substances that contain Hg^2+^ and have produced an extraordinarily low discharge standard [6,7]. For example, the US EPA standard is approximately 200pM [8], Japan’s standard is 40 ng/g, and China’s is 0.05 μg/L. However, current Hg^2+^ detection methods face numerous challenges as a result of the low discharge limit. The traditional technologies that detect Hg^2+^ are atomic fluorescence spectroscopy (AFS), electrochemistry (EC), and surface plasmon resonance (SPR) [9,10,11,12,13,14,15,16]. Thus, very sensitive pollution detection for Hg^2+^ is necessary [17,18]. The commonly used SPR optical sensing technologies have several advantages, for example, simple operation, low cost, and good stability [19,20,21]. Recently, graphene and MoS_2_ based SPR sensors were used as sensing layer materials [22,23,24,25,26]. However, due to the limited availability or shortage of sensing materials on gold surfaces, the use of old-fashioned SPR techniques for detecting low concentrations of metal ions is a big challenge. It is important to address the trace mercury ion low adsorption on to the current available sensing materials such as graphene, etc. [27,28]. Hence, recent research directions have focused on how to detect trace mercury ions and have illustrated the TaS_2_-based SPR sensors sensing mechanism.

The two-dimensional (2D) TaS_2_ material is currently a hot topic for researchers after the discovery of monolayer 1T-TaS_2_ [29,30]. TaS_2_ has a substantial potential for the removal of Hg^2+^, which has made it a perfect candidate for Hg^2+^ sensing [31,32,33,34,35]. By selecting the TaS_2_ material for a sensing layer as shown in Figure 1, a sensitive TaS_2_-based SPR biosensor was successfully fabricated to detect Hg^2+^, with TaS_2_ nanosheets as the sensing layer. This novel SPR sensor is compatible with the demands of trace Hg^2+^ detection, and these TaS_2_ based biosensors detection limit approaches 1 pM. It is evident from the results that the sufficient unsaturated bonds in the TaS_2_ structure and large interlayer spacing-based biosensors promote adsorption of Hg^2+^ and efficient transport.

## 2. Experiment

### 2.1. Synthesis of Superstructure TaS_2_ Nanoflakes

The liquid phase stripping method was used to produce the 2D TaS_2_ material ethanol suspensions. Using ultrasound and a mixture of purified water, the TaS_2_ interlayer van der Waals force was overcome. Using a centrifuge, TaS_2_ dispersions were formed by removing the TaS_2_ block from single and multilayer TaS_2_. First, 20 mg TaS_2_ was dissolved in 20 mL water, and we treated the entire solution in bath sonication at a power up to 300 W for 10 h. Furthermore, we controlled the effective temperature of the bath sonication at approximately 10 °C. After the entire procedure, the acquired TaS_2_ suspension was centrifuged for one hour with a rotation rate of 5000 r/min, then left for 12h for natural settling. For the upper layer liquid using a rotation rate of 12,000 r/min, the final step was centrifuged for 40 min. The subsequent supernatant was made of several layers of TaS_2_ dispersion for use in the next experiments.

### 2.2. Assembling of TaS_2_ Nanoflakes

TaS_2_ nanoparticles accumulated on gold (Au) chips using a layer-by-layer technique. TaS_2_ nanoparticle solution and polyetherimide (PEI) alternately flowed past the cell, and finally the TaS_2_ layers were assembled by electrostatic adsorption. Surplus cations were washed by Milli-Q de-ionized water (18.2 MΩ cm) flowing past the cell. In each dipping period, the thickness increased, and the liquid current velocity was controlled to be lower than 1 mL/min to avoid separation of the finished films.

### 2.3. Mercury Ion Detection

Mercury ion in a water solution at concentrations from 10^−12^ M to 10^−6^ M as imitations of domestic water supply were prepared to systematically pass through SPR biosensor equipment. The buffer was domestic water. We washed the sample using ultra-pure water after immersing it in Hg^2+^ solutions for half an hour, and then spectrum signals were recorded. The detection signal was exported through a computer linked with a Kretschmann structure electrochemical in situ time resolved SPR sensor.

## 3. Results and Discussion

### 3.1. Numerical Simulation

The monolayer TaS_2_ thickness is 0.361 nm [36,37]. The incident wavelength is 633 nm. In order to simulate this sensor sensitivity, the transmission matrix method is used. Type BK7 glass was selected as coupling prism, and 50 nm Au film was chosen as the metal layer. Equations for the BK7 glass refractive index was obtained from references [38,39]. The Au refractive index is represented by the Drude–Lorentz model [39]. The Au film surface was covered by TaS_2_ to prevent Au from being oxidized in order to further enhance the sensibility of the designed sensors. The refractive index of TaS_2_ is 12.25 + 3.06i [40,41,42]. The sensitive medium refractive index is expressed as n_s_ = 1.33 + Δn, whereas Δn designates the sensitive medium refractive index change caused by biological and chemical reaction.

The transfer matrix method was used [43] to calculate the N-layer model incident TM-polarized light reflectivity. These layers are piled vertically to the direction of the BK7 glass coupling prism for these TaS_2_ allotrope sensors, and the name of each layer is selected using the dielectric constant (ε_k_), thickness (d_k_) and refractive index (n_k_), separately.
(1)$$\left[\begin{array}{c} U_{1}\\ V_{1} \end{array}\right]=M\left[\begin{array}{c} U_{N-1}\\ V_{N-1} \end{array}\right]$$
(2)$$\prod {}_{K=2}^{N-1}M_{K}=\left[\begin{array}{cc} M_{11} & M_{12}\\ M_{21} & M_{22} \end{array}\right]$$
(3)$$M_{K}=\left[\begin{array}{cc} \cos\beta_{K} & \frac{-\sin\beta_{K}}{q_{k}}\\ -{\mathrm{iq}}_{k}\sin\beta_{k} & \cos\beta_{k} \end{array}\right]$$
(4)$$r_{p}=\frac{\left(M_{11}+M_{12}q_{5}\right)q_{1}-\left(M_{21}+M_{22}q_{5}\right)}{\left(M_{11}+M_{12}q_{5}\right)q_{1}+\left(M_{21}+M_{22}q_{5}\right)}$$
(5)$$R_{p}={\left|r_{p}\right|}^{2}$$

The transformation of the sensing medium might change the resonance angle (Δθ), where S_R1_ = Δθ/Δn represents the sensitivity [44]. The BK7 glass has a low refractive index coupling prism to use in these biosensors. Figure 1 shows a conventional Kretschmann structure SPR biosensor that has a metallic layer to activate SPP. By applying a BK7 glass prism, the resulting Au metal layer SPR biosensor sensitivity is 133°/RIU, as shown in Figure 2. For a biochemical sensor, this sensitivity is unsatisfied [45]. Utilizing 2D materials TMDCs TaS_2_, we have developed SPR biochemical sensors in the current paper to enhance the sensitivity of the sensors [46,47,48].

Figure 2a–e shows the changes of reflectivity versus the incident angle. The resonance angles of the TaS_2_ sensor shift to higher incidence angles at one layer and the sensitive media refractive indices increase. Sensitivity can be increased by using 2D TMDCs material TaS_2_-based biochemical sensors. Figure 2f shows the changes in sensitivity of TaS_2_ on the Au surface. It shows that the biosensor sensitivity increases and then decreases with the addition of TaS_2_ layers from 1 to 9; the highest sensitivity for TaS_2_ biosensors is 201°/RIU when the TaS_2_ layers accumulate to 8 layers. The most likely reason for the transformation is because the light utilization rate reduces with an increase in the number of TaS_2_ layers. Therefore, the number of TaS_2_ layers has an optimal value and cannot be increased unboundedly.

From Table 1, we can see that TaS_2_-based Kretschmann structure SPR biosensors show an obviously improvement.

### 3.2. Application of TaS_2_ Based SPR Biosensors in Heavy Metal Ion Detection

The simulation results discussed in the last section provide details for depositing TaS_2_ nanoparticles onto SPR chip surfaces. First, using layer-by-layer (LBL) techniques, we assembled the as-prepared TaS_2_ nanosheets onto the SPR chip Au surface. Subsequently, using scanning electron microscopy (SEM), the surface morphology of TaS_2_ nanosheets over the SPR chips was characterized, as shown in Figure 3. By increasing the dipping cycle, TaS_2_ nanosheets were more thickly deposited, as shown in Figure 3a–c. Unexpectedly, it was observed that the morphology of the TaS_2_ nanosheets was not affected by the LBL process, as can be seen in the right side of Figure 3a–c. Thus, assembly of TaS_2_ nanosheets over SPR chip surfaces can be tuned by LBL assembly techniques. We have obtained SPR curves through different layers as the superstructure TaS_2_ materials were assembled via LBL method.

We have prepared SPR sensors with chips coated with TaS_2_ nanosheets that, using SEM, can be seen to detect heavy metal ions. SPR spectra for different mercury ion concentrations from 10^−12^ M to 10^−6^ M were obtained, as shown in Figure 4a. We have identified the obvious shift curve toward higher resonance, which shows the Hg^2+^ adsorption onto the sensing layer material. The Hg^2+^ detection limit value in the calibration curve for the SPR angle shift versus Hg^2+^ concentration, from 10^−12^ M to 10^−6^ M Hg^2+^ solution concentration, 1 pM, were found to be the SPR sensor detection range [55]. This method’s detection limit was 200 times lower compared with other techniques [56,57,58]. Figure 4b displays the SPR sensors calibration curve of angle shift along with mercury ion concentration without TaS_2_ material, which is also in the range of 10^−12^–10^−6^, and the SPR spectra do not have much diversification. Figure 4c shows the incident angle changes for mercury ion concentrations from 10^−12^ M to 10^−6^ M; after linear fitting, we get a sensitivity of 10.36°/μM.

Apart from detecting Hg^2+^ in everyday drinking water, we also calculated the SPR sensor selectivity through investigating their responses to other heavy metal ions, for instance, Cr^3+^, Cr^6+^, Ag^2+^, Pb^2+^, Cu^2+^, and Cd^2+^. Under the same conditions, the response of the SPR sensors to dissimilar interfering ions at a level of 10^−6^ M concentration is shown in Figure 4d. In order to prevent the interference of an ion, we detected six different salts using the same concentration. The signal response ∆θ of the six cations were similarly low, nearly 100 mdeg on the same 10^−6^ M concentrations. The results reveal that the SPR sensors have a strong selectivity to mercury ions compared with the other metal ions.

### 3.3. Study of the Mechanism of High Sensitivity of These SPR Biosensors

The mechanism of the sensitivities of these TaS-based SPR sensors to detect Hg^2+^ was measured using elemental mapping of TaS_2_ nanoflakes. Shown in the Figure 5, the elemental mapping characterizations of superstructure TaS_2_ nanosheets by STEM EDS are displayed. Figure 5a shows a STEM image of TaS_2_ nanoparticles; in the same way Figure 5b–d reveal Ta, S, and Ta + S EDS elemental maps. The distance between the layers of TaS_2_ nanosheets is large, which is favourable for Hg^2+^ to enter the inner space where S atoms as binding sites are most likely located. These features allow the use of these TaS_2_ nanosheets to detect Hg^2+^ in SPR sensors.

To get a clear picture of these highly sensitivity SPR sensors, we used the ESCALAB-250 instrument to measure X-ray photoelectron spectroscopy (XPS) in order to analyse the chemical states of Ta and S in structured TaS_2_ nanoparticles and the original TaS_2_ block. From Figure 6a, it is clear that two characteristic peaks of 167.1 eV and 168.4 eV are accredited to the 2p^2/3^ and 2p^1/2^ energy levels of divalent sulphide ions. Figure 6b, on the left side, illustrates that, at low intensity, the peak at 163.7 eV may be attributed to bridging S^2-^, which represents unsaturated S atoms. The Ta 3d spectra of the TaS_2_ powder are demonstrated by the peaks at 404.5 eV and 466.3 eV, corresponding to binding energies of Ta 4p^3/2^ and Ta 4d^1/2^, respectively (Figure 6a,b), which suggests the presence of TaS_2_ structure 422.3 eV is the loss of Ta. 

The left side of Figure 6b shows new peaks, which suggests the occurrence of the TaS_2_ structure. The atomic configuration of TaS_2_ is very dense on the surface of the substrate and also has a very high electronic conductivity, thus facilitating the combination of Hg^2+^. In TaS_2_ nanoflakes, the increased sensitivity of Hg^2+^ in the SPR sensor is promoted by the unsaturated S atoms. 

## 4. Conclusions

Innovative SPR biochemical sensors using 2D TMDCs TaS_2_ were designed and constructed to enhance the sensitivity to Hg^2+^ ions. The proposed design has improved the sensitivity of biochemical sensors to 201°/RIU for TaS_2_. The SPR sensor was developed, and its performance was evaluated through a combination of simulation and experimental results. The detection sensitivity limit of TaS_2_-based SPR biosensors is nearly 1 pM. TEM, HRTEM, and XPS were used to investigate the mechanism of these highly sensitive SPR sensors. From the obtained results, having a large inter-layer spacing and sufficient non-structural combination in the proposed TaS_2_-based structure makes Hg^2+^ effectively transported and absorbed. These studies provide efficient methods for the detection of low Hg^2+^ concentrations and generalize the utilization of structured TaS_2_ nanosheets.

## Figures and Tables

**Figure 1 nanomaterials-12-02075-f001:**
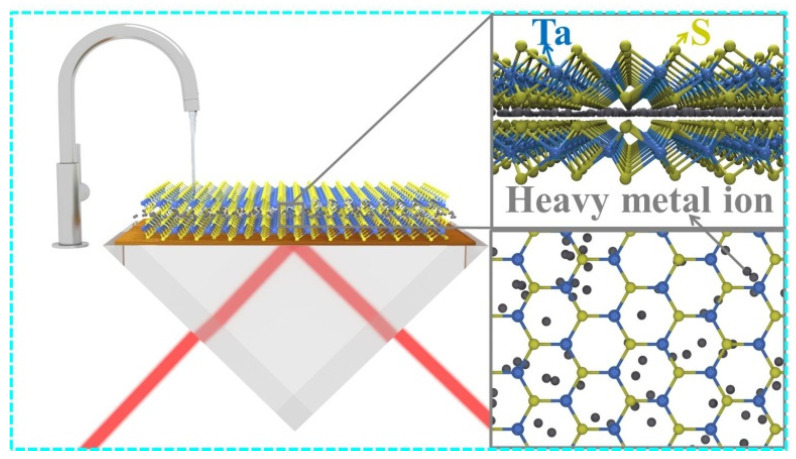
TaS2 nanoflakes as the sensing layer to detect trace mercury ions in SPR sensors.

**Figure 2 nanomaterials-12-02075-f002:**
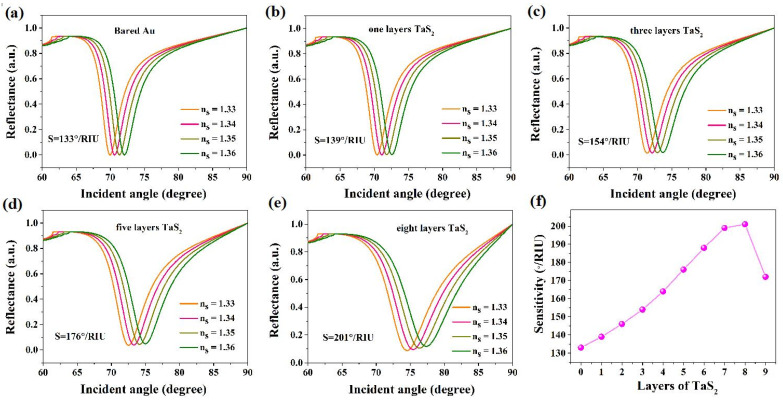
(**a**–**e**) Simulation change of reflectivity relative to incident angles with different numbers of TaS_2_ nanosheets; (**f**) variation of sensitivity with respect to the number of TaS_2_ nanosheets.

**Figure 3 nanomaterials-12-02075-f003:**
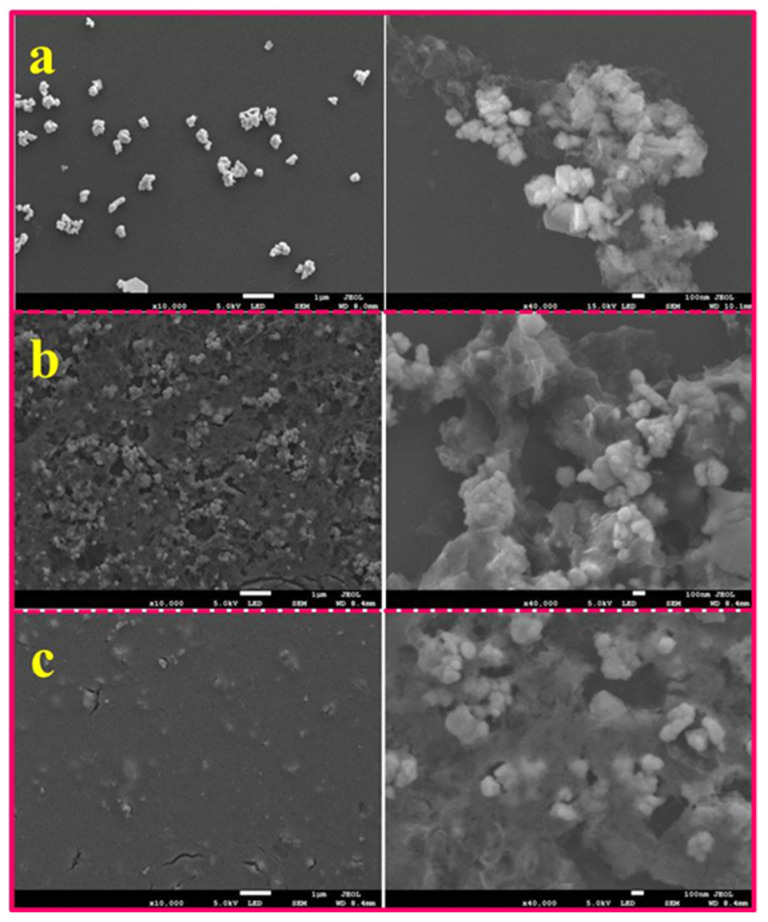
(**a**–**c**) The SEM images of the superstructure TaS_2_ nanoparticles assembled with different thicknesses were obtained for 1 μm scale bar on the SPR chip surface. The panels on the right side are the magnified images, with a scale bar of 100 nm.

**Figure 4 nanomaterials-12-02075-f004:**
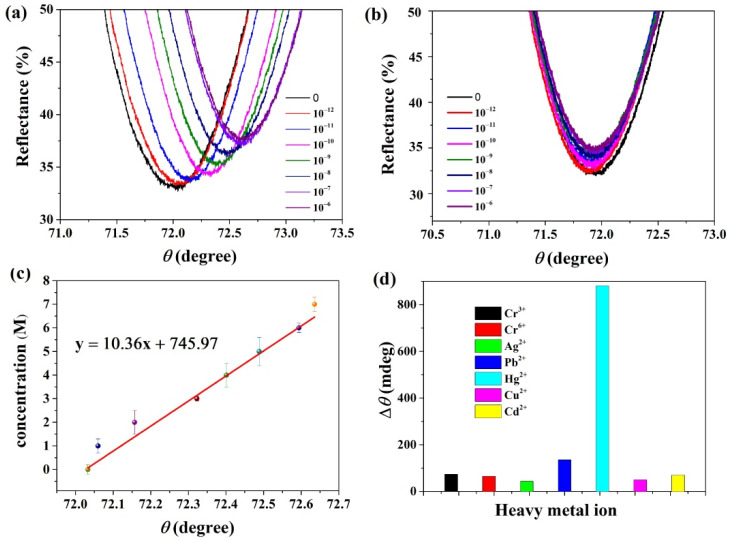
(**a**) The TaS_2_ SPR spectra at different concentrations of Hg^2+^ from 10^−12^ to 10^−6^ M; (**b**) SPR spectra without TaS_2_ at different concentrations of Hg^2+^ from 10^−12^ to 10^−6^ M; (**c**) the fitting curve of the incident angle at different concentrations of Hg^2+^ from 10^−12^ to 10^−6^ M; (**d**) selectivity of these TaS_2_ based Hg^2+^ sensors.

**Figure 5 nanomaterials-12-02075-f005:**
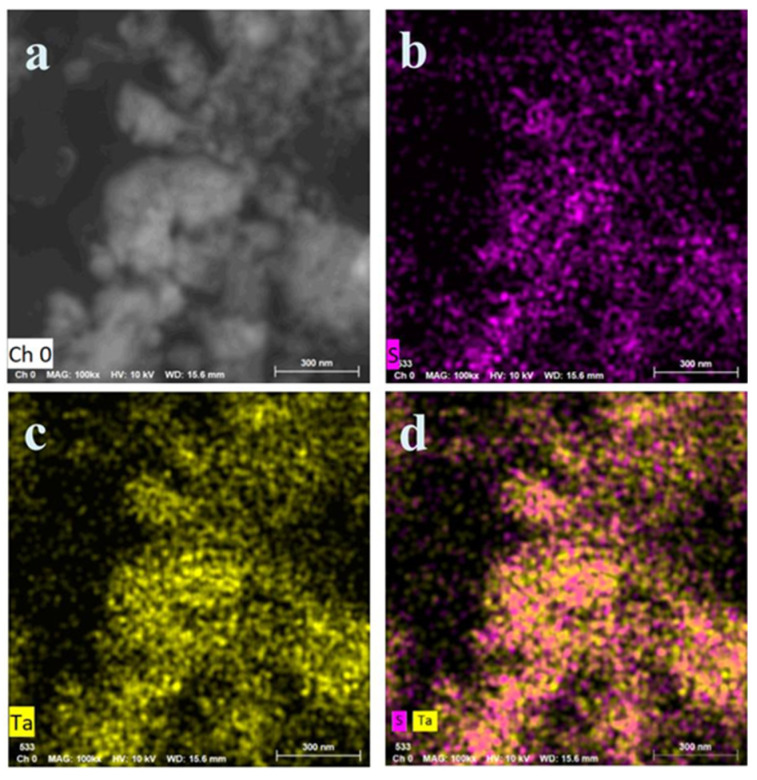
Elemental mapping characterizations of superstructure TaS_2_ nanosheets by STEM EDS. (**a**) STEM image of TaS_2_ nanosheets, similarly acquired with (**b**–**d**) Ta, S, and Ta + S EDS elemental maps.

**Figure 6 nanomaterials-12-02075-f006:**
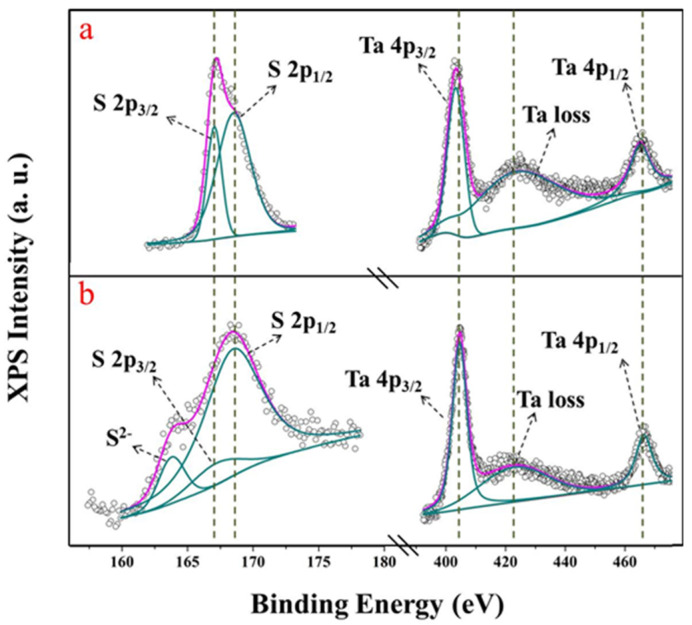
(**a**) XPS spectra of bulk TaS_2_; (**b**) XPS spectra of TaS_2_ nanoflakes.

**Table 1 nanomaterials-12-02075-t001:** Different 2D material sensing medium layers to improve biosensors.

2D Materials	Optimal Sensitivity	Maximum Sensitivity	Detection Limit	Ref.
Graphene oxide	91.1°/RIU	2715 nm/RIU	10^−11^ mol/L	[49,50]
MoS_2_	49.2°/RIU	0.64 μA/ppb	11.52 × 10^−3^ ppb	[51,52]
MoSe_2_	50.4°/RIU	2524 nm/RIU	3.5 nM	[53]
WS_2_	48.6°/RIU	2459 nm/RIU	3.3 nM	[54]
TaS_2_	201°/RIU	10.36°/μM	1 pM	Our work

## Data Availability

The data that support the findings of this study are available from the corresponding author upon reasonable request.

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
