# Peer review of "High Quality TaS2 Nanosheet SPR Biosensors Improved Sensitivity and the Experimental Demonstration for the Detection of Hg2+"

_nanomaterials, 2022, doi:10.3390/nano12122075_

Round 1

Reviewer 1 Report

Review of Yu Jia and al. Nanomaterials 2022

The major problem of this manuscript is the English language and style which makes the reading of the manuscript very complicated and leave some conclusions and interpretations unclear. As it is, I believe that this manuscript cannot be accepted in Nanomaterials as it is simply not possible to do an in-depth review because of the style.

There are also some major scientific points that must be addressed before resubmitting:

  • In overall, I am not convinced by the SPR simulations shown in Fig.2. Clearly, and as stated by the author at the end of the introduction, the main advantage in term of sensing brought by these multiple TaS2 layers, is to maximise the adsorption of mercury ions inside the plasmonic evanescent wave. For this reason, measuring the LOD of mercury ions for different numbers of TaS2 layers (in average over the biochips, using the layer-by-layer technique) would have been a better approach and would also be a direct demonstration of this statement. In comparison, the gain in sensitivity shown by the authors for purely plasmonic reasons seems quite anecdotical and not very convincing. First, the incident wavelength is not given. Depending on the working wavelength, a 50 nm gold film is not the optimum in term of sensitivity and should not be taken as the reference SPR sensor (this would explain the quite low sensitivity of 133°/RIU for the bare gold film). I also do not understand the explanation for the sharp drop of sensitivity between 8 and 9 layer (“..the light utilization rate decreases with the raise of TaS2 layers”) and it looks more like a numerical artefact (the 9 layers reflectance curves are not shown). A small but important remark: its highly probable than the value of 12.25+3.06i given for TaS2 is the permittivity and not the index of refraction as such a huge real part for the index would completely kill the plasmon with only one or two layer of TaS2. Finally, I am not convinced that detailing the equations for the transfer matrix methods is necessary as it is a common tool in electromagnetism.
  • Table 1 shows the LOD of 5 different systems in the literature and all of them are expressed in different units (mol/L, ppb, pM,…). This makes the comparison difficult and this table much less useful.
  • Many important details are missing in the experimental section: the estimated average number of TaS2 layer used in the Hg detection is not given. From the resonance angle in figure 4 (72°) it looks like 5 layers were used which is strange as the authors stated that 8 was the optimum. The buffer used for testing the different ions is not given which is problematic as pH and counter ion is critical in term of solubility and stability when working with metal ions solutions. Fig 4d is interesting but have been done at high concentration (10-6). Is the selectivity still as good when working closer to the LOD?
  • Fig 7 is lacking some explanation on how it was obtained and what it represents (I suppose ‘lateral distance’ means distance from the gold interface? why a ‘normalized’ electric field is in 107 V/m unit ?)  But I fail to understand the result: TaS2 layers are supposed to be 0.361 nm thick. Even 10 layers is only a few nm: how can the plasmonic field reach a maximum at 150 nm from the gold surface ? 

In overall, I believe this manuscript needs an important rework in term of style and scientific content before considering a publication.

Author Response

Reviewer 1

The major problem of this manuscript is the English language and style which makes the reading of the manuscript very complicated and leave some conclusions and interpretations unclear. As it is, I believe that this manuscript cannot be accepted in Nanomaterials as it is simply not possible to do an in-depth review because of the style.

There are also some major scientific points that must be addressed before resubmitting:

In overall, I am not convinced by the SPR simulations shown in Fig.2. Clearly, and as stated by the author at the end of the introduction, the main advantage in term of sensing brought by these multiple TaS2 layers, is to maximise the adsorption of mercury ions inside the plasmonic evanescent wave. For this reason, measuring the LOD of mercury ions for different numbers of TaS2 layers (in average over the biochips, using the layer-by-layer technique) would have been a better approach and would also be a direct demonstration of this statement. In comparison, the gain in sensitivity shown by the authors for purely plasmonic reasons seems quite anecdotical and not very convincing. First, the incident wavelength is not given. Depending on the working wavelength, a 50 nm gold film is not the optimum in term of sensitivity and should not be taken as the reference SPR sensor (this would explain the quite low sensitivity of 133°/RIU for the bare gold film). I also do not understand the explanation for the sharp drop of sensitivity between 8 and 9 layer (“..the light utilization rate decreases with the raise of TaS2 layers”) and it looks more like a numerical artefact (the 9 layers reflectance curves are not shown). A small but important remark: its highly probable than the value of 12.25+3.06i given for TaS2 is the permittivity and not the index of refraction as such a huge real part for the index would completely kill the plasmon with only one or two layer of TaS2. Finally, I am not convinced that detailing the equations for the transfer matrix methods is necessary as it is a common tool in electromagnetism.

Answer: I rewrote the abstract as: “The structural characterizations and simulation data show that the SPR sensor with high sensitivity can be obtained as a result of sufficient unstructured bonds in the TaS2 structure as well as the large inter-layer spacing which highly supports transporting and absorbing the mercury ions. In this paper, the role of TaS2 nanoparticles in SPR sensor was explored by simulation and experiment, and the TaS2 layer in SPR sensor was characterized by SEM, Elemental Mapping, XPS and other methods. The application range of structure TaS2 nanoparticles is explored, these TaS2 based sensors were applied to detect Hg2+ ions and detection limit approach 1 pM, and an innovative idea for designing highly sensitive detection techniques is provided.”

The incident wavelength is 633nm. The sensitivity has the optimum number.

Table 1 shows the LOD of 5 different systems in the literature and all of them are expressed in different units (mol/L, ppb, pM,…). This makes the comparison difficult and this table much less useful.

Answer: Although the results in Table 1 are not in the unit cause the different measure method. Table 1 gives the appropriate number for readers to know.

Many important details are missing in the experimental section: the estimated average number of TaS2 layer used in the Hg detection is not given. From the resonance angle in figure 4 (72°) it looks like 5 layers were used which is strange as the authors stated that 8 was the optimum. The buffer used for testing the different ions is not given which is problematic as pH and counter ion is critical in term of solubility and stability when working with metal ions solutions. Fig 4d is interesting but have been done at high concentration (10-6). Is the selectivity still as good when working closer to the LOD?

Answer: Simulation results are idealizing datum. The actual experiments used simulation results as reference to performance, but always can’t same as simulation results. So this paper also is direct and example for how to detect heavy metals in water using SPR sensors. “The buffer is domestic water.” is emphasized in part 2.3 line 2.

Fig 7 is lacking some explanation on how it was obtained and what it represents (I suppose ‘lateral distance’ means distance from the gold interface? why a ‘normalized’ electric field is in 107 V/m unit?)  But I fail to understand the result: TaS2 layers are supposed to be 0.361 nm thick. Even 10 layers is only a few nm: how can the plasmonic field reach a maximum at 150 nm from the gold surface?

In overall, I believe this manuscript needs an important rework in term of style and scientific content before considering a publication.

Answer: I deleted Fig. 7.

Reviewer 2 Report

In this paper the authors present a SPR sensor to detect Hg ions at low concentration. The platform is based on the use of TaS2 nanosheet on a Au film and a standard SPR configuration. With respect to the simple Au film, the Au/TaS2 system is able to increase the sensitivity and also seems to be selective towards Hg ions with respect to others. 

the idea to use 2D materials flakes to improve the performances of SPR sensors towards specific analytes is not new, but the use for Hg ions with TaS2 has not been reported with this sensitivity level.

The contents are interesting, but the manuscript needs significant revision before to be accepted.

First of all, the english grammar are rather bad and several sentences are completely wrong or impossible to be read, major examples:

  • the abstract: it sounds quite confuse and do not resume the contents of the paper correctly
  • pag 3 . section 3.1 line 4: "Equation that the BK7 glass refractive index is obtained from [34, 35]."
  • page 4: "For biochemical sensor, this sensitivity is unsatisfied [41]."...
  • page 6. line 1. "We have prepared the SPR sensors whose chips coated with TaS2 nanosheets which is a while film can be seen from SEM..."
  • page 6 "1 pM are found to be the SPR sensor detection range"....detection range or limit?
  • page 8 "From left picture of Fig. 6b shows new"....

From the technical point of view there are several points to be better discussed:

  1. from fig. 3 it is not clear how uniform are the deposition of TaS2 sheets. the SEM shows particles and bad uniform films. considering that the sensitivity is strongly dependent on the number of layers it seems here that the authors cannot control it
  2. the discussion on the selectivity towards Hg ions with respect to the other ions is not convincing. the TEM, EDS and XPS characterizations show something but did not analyze the direct interaction with the ions and the possible chemistry of these interaction. A better discussion is needed.

Being a paper on the SPR technology, I typically recommend to cite the top performers in terms of sensitivity:

Nat. Mater., 2009, 8, 867.

Nat. Mater., 2016, 15, 621–627.

Nanoscale Horiz., 2019, 4, 1153-1157

Photonics Research 2022, 10, 84-95

Author Response

Reviewer 2

In this paper the authors present a SPR sensor to detect Hg ions at low concentration. The platform is based on the use of TaS2 nanosheet on a Au film and a standard SPR configuration. With respect to the simple Au film, the Au/TaS2 system is able to increase the sensitivity and also seems to be selective towards Hg ions with respect to others. The idea to use 2D materials flakes to improve the performances of SPR sensors towards specific analytes is not new, but the use for Hg ions with TaS2 has not been reported with this sensitivity level. The contents are interesting, but the manuscript needs significant revision before to be accepted.

First of all, the english grammar are rather bad and several sentences are completely wrong or impossible to be read, major examples: the abstract: it sounds quite confuse and do not resume the contents of the paper correctly

pag 3 . section 3.1 line 4: "Equation that the BK7 glass refractive index is obtained from [34, 35]."

page 4: "For biochemical sensor, this sensitivity is unsatisfied [41]."...

page 6. line 1. "We have prepared the SPR sensors whose chips coated with TaS2 nanosheets which is a while film can be seen from SEM..."

page 6 "1 pM are found to be the SPR sensor detection range"....detection range or limit?

page 8 "From left picture of Fig. 6b shows new"....

Answer: I modified the writing in the manuscript. I rewrote the abstract as: “The structural characterizations and simulation data show that the SPR sensor with high sensitivity can be obtained as a result of sufficient unstructured bonds in the TaS2 structure as well as the large inter-layer spacing which highly supports transporting and absorbing the mercury ions. In this paper, the role of TaS2 nanoparticles in SPR sensor was explored by simulation and experiment, and the TaS2 layer in SPR sensor was characterized by SEM, Elemental Mapping, XPS and other methods. The application range of structure TaS2 nanoparticles is explored, these TaS2 based sensors were applied to detect Hg2+ ions and detection limit approach 1 pM, and an innovative idea for designing highly sensitive detection techniques is provided.”

I modified the English grammer.

From the technical point of view there are several points to be better discussed: from fig. 3 it is not clear how uniform are the deposition of TaS2 sheets. The SEM shows particles and bad uniform films. Considering that the sensitivity is strongly dependent on the number of layers it seems here that the authors cannot control it the discussion on the selectivity towards Hg ions with respect to the other ions is not convincing. The TEM, EDS and XPS characterizations show something but did not analyze the direct interaction with the ions and the possible chemistry of these interaction. A better discussion is needed.

Being a paper on the SPR technology, I typically recommend to cite the top performers in terms of sensitivity: Nat. Mater., 2009, 8, 867. Nat. Mater., 2016, 15, 621–627. Nanoscale Horiz., 2019, 4, 1153-1157. Photonics Research 2022, 10, 84-95

Answer: I quoted the reference.

[13] A.V. Kabashin, P. Evans, S. Pastkovsky, W. Hendren, G.A. Wurtz, R. Atkinson, R. Pollard, V.A. Podolskiy, A.V. Zayats, Plasmonic nanorod metamaterials for biosensing, Nat. Mater., 8 (2009) 867-871.

[14] K.V. Sreekanth, Y. Alapan, M. ElKabbash, E. Ilker, M. Hinczewski, U.A. Gurkan, A. De Luca, G. Strangi, Extreme sensitivity biosensing platform based on hyperbolic metamaterials, Nat. Mater., 15 (2016) 621-627.

[15] D. Garoli, E. Calandrini, G. Giovannini, A. Hubarevich, V. Caligiuri, F. De Angelis, Nanoporous gold metamaterials for high sensitivity plasmonic sensing, Nanoscale Horizons, 4 (2019) 1153-1157.

[16] R. Yan, T. Wang, X. Yue, H. Wang, Y.-H. Zhang, P. Xu, L. Wang, Y. Wang, J. Zhang, Highly sensitive plasmonic nanorod hyperbolic metamaterial biosensor, Photon. Res., 10 (2022) 84-95.

Round 2

Reviewer 2 Report

The new version of the manuscript has been only slightly improved.

there are still a lot of english grammar errors along the text that made it really hard to be understood

it is not clear why the authors mentions TaS2 nanoparticles....is it not a flakes or layers? from figure 3 it seems that the deposition of TaS2 is not well controlled in term of thickness and uniformity

I really recommend a full revision with the support of an english writer. then the paper can be considered for publication

Author Response

nanomaterials-1691979 Title: High Quality TaS2 Nanosheet SPR Biosensors Improved Sensitivity and the Experimental Demonstration for the Detection of Hg2+ Author: Yue Jia et al. Dear reviewers and editors This is the revised manuscript of original manuscript (High Quality TaS2 Nanosheet SPR Biosensors Improved Sensitivity and the Experimental Demonstration for the Detection of Hg2+). We appreciate that the editors can give us a chance to resubmit our manuscript. We really cherish this opportunity and try our best to revise this paper according to the review comments. First of all, please convey our best regards to the reviewers. They have put forward some constructive and informative advice and suggestions for improving this manuscript. The advice and suggestions are of a great help to our paper. These comments are all valuable and very helpful for revising and improving our paper, as well as the important guiding significance of our research. We have studied comments carefully and have made corrections which we hope meet with approval. We prepared this cover letter which has a detailed response to reviewers’ comments. We first copy the comment of the reviewers and then give the reply immediately after that. Your further suggestion and comments will be greatly appreciated. We are earnestly looking forward to your decision. Thanks very much for your attention to our paper. Best regards! Corresponding author: Houzhi Cai E-mail addresses: hzcai@szu.edu.cn Response to Reviewer 1 Comments The new version of the manuscript has been only slightly improved. There are still a lot of english grammar errors along the text that made it really hard to be understood It is not clear why the authors mentions TaS2 nanoparticles....is it not a flakes or layers? from figure 3 it seems that the deposition of TaS2 is not well controlled in term of thickness and uniformity I really recommend a full revision with the support of an english writer. Then the paper can be considered for publication Response: Thank you so much for review my paper. I have checked and modified the English writing. We assembled the as-prepared TaS2 nanosheets onto the SPR chip Au surface. The TaS2 nanosheets solution is uniforma dispersion, after deposited on the Au surface the solvent will reduce to pile up the TaS2 nanosheets to form layer.

Round 3

Reviewer 2 Report

The manuscript is now ok for publication